# Systemic Effects of Photoactivated 5,10,15,20-tetrakis(*N*-methylpyridinium-3-yl) Porphyrin on Healthy *Drosophila melanogaster*

**DOI:** 10.3390/biotech13030023

**Published:** 2024-07-03

**Authors:** Ana Filošević Vujnović, Sara Čabrijan, Martina Mušković, Nela Malatesti, Rozi Andretić Waldowski

**Affiliations:** Faculty of Biotechnology and Drug Development, University of Rijeka, Radmile Matejčić 2, 51000 Rijeka, Croatia; sara.cabrijan@biotech.uniri.hr (S.Č.); martina.muskovic@biotech.uniri.hr (M.M.); nela.malatesti@biotech.uniri.hr (N.M.); randretic@biotech.uniri.hr (R.A.W.)

**Keywords:** photodynamic therapy, photosensitizing agent, 5,10,15,20-tetrakis(*N*-methylpyridinium-3-yl)-porphyrin, hydrogen peroxide, negative geotaxis, *Drosophila melanogaster*

## Abstract

Porphyrins are frequently employed in photodynamic therapy (PDT), a non-invasive technique primarily utilized to treat subcutaneous cancers, as photosensitizing agents (PAs). The development of a new PA with improved tissue selectivity and efficacy is crucial for expanding the application of PDT for the management of diverse cancers. We investigated the systemic effects of 5,10,15,20-tetrakis(*N*-methylpyridinium-3-yl)-porphyrin (TMPyP3) using *Drosophila melanogaster* adult males. We established the oral administration schedule and demonstrated that TMPyP3 was absorbed and stored higher in neuronal than in non-neuronal extracts. Twenty-four hours after oral TMPyP3 photoactivation, the quantity of hydrogen peroxide (H_2_O_2_) increased, but exclusively in the head extracts. Regardless of photoactivation, TMPyP3 resulted in a reduced concentration of H_2_O_2_ after 7 days, and this was linked with a decreased capacity to climb, as indicated by negative geotaxis. The findings imply that systemic TMPyP3 therapy may disrupt redox regulation, impairing cellular signaling and behavioral outcomes in the process. To determine the disruptive effect of porphyrins on redox homeostasis, its duration, and the mechanistic variations in retention across various tissues, more research is required.

## 1. Introduction

Photodynamic therapy (PDT) is a minimally invasive medical treatment that uses a combination of a photosensitizing agent (PA) and light of a specific wavelength to selectively destroy cancer cells and/or other abnormal tissues [1,2,3,4]. Light absorption by the PA molecule leads to an excited singlet state, which, after an intersystem crossing, leads the PA into the triplet excited state. From its triplet excited state, the PA can transfer energy directly to nearby oxygen molecules (type II process), or through electron/hydrogen atom transfer (type I), react with surrounding biomolecules, leading to a localized increase in the reactive oxygen species (ROS) that damage cancer cells [5]. The PA is mostly administered locally to the affected area in the treatment of skin conditions associated with tumors [6]. New approaches have been developed for the use of PDT in systemic treatments of tumors that are difficult to operate on, as well as in metastatic disease [7,8]. For further advances in PDT, it is important to optimize PA dosage and exposure time and collect information about the systemic side effects of the PA that is retained in the tissue after the treatment.

One class of commonly used PAs are porphyrins [9], and the challenge in the PDT field is to develop new classes of porphyrin derivatives with high specificity for internalization into cancerous cells, where they will activate mechanisms of cell death and result in antimetastatic activity. In animal and cell culture studies, cationic 5,10,15,20-tetra-(*N*-methyl-4-pyridyl) porphyrin (TMPyP4) has shown promising effects in the treatment of cervical cancer [10] and ovarian carcinoma [11], due to ROS-mediated mitochondrial dysfunction and DNA damage. These effects can be enhanced by adding hydrogen peroxide (H_2_O_2_) to the cell culture, demonstrating that a high-ROS microenvironment present in the cancerous cells plays a role in chemotherapeutic drug action [12]. Cationic 5,10,15,20-tetrakis(*N*-methylpyridinium-3-yl)-porphyrin (TMPyP3) is a positional isomer of TMPyP4 with similar chemical properties (Figure 1). The addition of a long alkyl chain to tripyridylporphyrin resulted in antiviral effects and provided a model for photodynamic antimicrobial chemotherapy [13], while *N*-methylated and *N*-oxidized tripyridylporphyrins with long alkyl chains have shown strong PDT effects and inhibitory activity in sphingolipid metabolism, suggesting a potential for the treatment of cancerous cells [14]. While these results are encouraging, it is important to note that they were obtained in cell cultures, leaving the specificity in and effectiveness on tumors in an in vivo model unknown.

Due to the low cost of husbandry and genetic versatility, *Drosophila melanogaster* presents an ideal laboratory organism that can be developed as a model for investigating the effectiveness of PDT, screening for the toxicity of the porphyrins and generating a description of genetic and cellular mechanism of action [15]. *Drosophila* has a high genetic similarity with humans, a low gene redundancy and similar cellular signaling pathways—advantages that contributed to its use as an in vivo model for screening compounds against different types of cancers. However, the testing efficiency and potential toxicity of different PAs using *Drosophila* as in vivo model for PDT have been limited [16,17]. Protoporphyrin-IX (PP-IX) was fed to *Drosophila* to determine the effect on the lifespan duration. PP-IX extended the lifespan in wild-type flies but shortened it in flies deficient in the superoxide dismutase, suggesting a dual effect, anti- or pro-oxidant, depending on the genotype of the flies [18]. Zinc(II)-centered porphyrin combined with ruthenium(II) polypyridyl groups was shown to be nontoxic in *Drosophila* larvae. Importantly, in the absence of light, this PA passed through cell membranes accumulating in the cytosol at lower concentrations and in the nucleus when administered at a higher concentration [19]. These findings show that there is a potential for *Drosophila* to be used for screening new PAs for PDT.

In this study, our primary objective was to analyze the systemic effect of TMPyP3 in healthy adult male *Drosophila* by assessing the potential impact on the nervous system and, consequently, behavior. Given that TMPyP3 is an organic compound that can react with the ingredients used to prepare food, we first optimized the nutrient medium for oral administration. We successfully established the most effective feeding method and the protocol for ensuring the highest ingestion of TMPyP3. We measured the amount of TMPyP3 retained in the body and the head up to one week after the administration. To evaluate the consequences of TMPyP3 presence in the body and the effect of the exposure to photosensitizing red light, we examined the levels of hydrogen peroxide (H_2_O_2_) in the head and the body one and seven days post-treatment. To determine the potential behavioral effects of TMPyP3 treatment, we measured the ability of flies to climb a vertical surface using a negative geotaxis test—a common approach for measuring consequences of oxidative stress, such as those associated with ageing and neurodegenerative disease.

## 2. Materials and Methods

### 2.1. TMPyP3 in Nutrient Media

5,10,15,20-tetrakis(*N*-methylpyridinium-3-yl)porphyrin (TMPyP3) was synthesized and characterized as described earlier [20]. As we planned to orally administer TMPyP3 to adult flies, the stability of TMPyP3 was first tested in three different *Drosophila* nutrient media after 24 h of incubation. Triplicates of M1, M2 and M3 with 40 μM TMPyP3 were prepared in the vials and kept in the dark. For each medium (M1, M2 and M3), we used a different recipe. In detail, 73 milliliters of tap water, 4.5 milliliters of molasses, 1.1 g of sucrose, 2.6 g of powder yeast, 0.9 g of type II agar, 0.75 milliliters of propionic acid (Sigma Aldrich, 99%), and 0.75 milliliters of 10% p-hydroxybenzoic acid methyl ester (NIPAGINE, Roth, 99%) dissolved in 95% ethanol (VWR) comprised Medium 1 (M1). Also, 75 milliliters of tap water, 1.5 g of type II agar, 3 g of sucrose, and 6.9 g of powder yeast made up Medium 2 (M2). Agar and sucrose without yeasts made up Medium 3 (M3) (75 mL tap water, 1.5 g type II agar, 3 g sucrose). We took a 25 mg of M1, M2, and M3, and each one dissolved in duplicate in distilled water until the final volume of 1 mL to determine TMPyP3 concentration. The samples were centrifuged for ten minutes at 15,000 rpm after being heated for thirty minutes at 77 °C. A 96-well plate was filled with 200 µL of each medium supernatant (including and excluding TMPyP3) in triplicate. TMPyP3’s fluorescence emission spectra were then captured using an excitation wavelength of 420 nm and emission in the range of 550–800 nm.

### 2.2. Light Source and Irradiation Conditions

The same LED-based source of red light was used as in the previously described work [20]. Starved flies fed with 40 µM TMPyP3 for 180 min in M3, and their non-starved counterparts fed with TMPyP3 for 24 h in the same concentration, were used to test the light-activated TMPyP3 in adult flies. The flies were exposed to red-light irradiation (λ_max_ = 645 nm, Δλ_FWHM_ = 24 nm, 40 mW/cm^2^) for 10 or 20 min (24 and 48 J/cm^2^ of light dose, respectively).

### 2.3. TMPyP3 Feeding Procedure

For the in vivo experiments, we used *Drosophila melanogaster* wild-type males of the *Canton-S* strain. First, 3–5 day-old flies raised on cornmeal food in the condition of 12 h light and 12 h dark, at 24 °C and 70% relative humidity, were anesthetized with CO_2_ before being transferred in groups of 10. Flies were starved on 1 g of type II agar dissolved in 100 mL of tap water for 16 h in an incubator in the dark at 24 °C and 70% relative humidity. Following starvation, flies were moved to M3 and exposed in the dark for 30, 90, or 180 min to 40 µM TMPyP3. The non-deprived fly group was housed in the identical conditions as the starved group for 24 h after being moved to M3 supplemented with 40 µM TMPyP3. As soon as the flies were fed, we prepared a whole-body homogenate to measure TMPyP3 in the entire body using lysis buffer, which is composed of 1 × PBS (phosphate-buffered saline) and 0.1% (*v*/*v*) Triton X-100 (Sigma Aldrich), at pH 7.4. The ratio of 300 µL of buffer to 5 mg of tissue sample was used to calculate the extraction buffer volume. The analytical grades of NaCl, KCl, Na_2_HPO_4_, and KH_2_PO_4_ were present. Following mechanical homogenization, the samples were centrifuged for 30 min at 4 °C at 14,000 rpm and then incubated for 20 min in an ice bath. Using a calibration curve for TMPyP3 in PBT, the amount of TMPyP3 in the tissue extracts was ascertained. The fluorescence emitted by the samples was measured using a Tecan Infinity m1000Pro microplate reader with a 420 nm excitation wavelength, detecting emissions at 710 nm.

### 2.4. TMPyP3 Ingestion and Retention in the Tissue after the Oral Administration

To assess the concentration of TMPyP3 retention following one and seven days, within either head or body extracts, we followed a consistent method of extracting and measuring TMPyP3 as described above. During this examination, 3–5-day-old flies underwent a 16 h starvation period followed by a 180 min exposure to 40 µM TMPyP3 on M3 in the absence of light (refer to Figure 2). Subsequent to the feeding process, the flies were either subjected to red-light emitting diodes for 10 min before being transferred to fresh M3 media, or directly shifted to fresh M3 and maintained on it for durations of one or seven days under constant darkness, preceding biochemical and behavioral assessments.

### 2.5. Quantification of the Hydrogen Peroxide (H_2_O_2_)

The 3–5-day-old flies underwent a 16 h period of fasting followed by 180 min of TMPyP3 ingestion under constant dark. Post-feeding, the flies were divided into two groups: one group was exposed to illumination at 40 mW/cm^2^ for 10 min (referred to as “Light” or “L”), while the other served as the “no light control” (referred to as “No Light” or “NL”). Following this, the flies were relocated to fresh M3 and allowed to remain in darkness for either one or seven days to prevent any potential impact of light on TMPyP3 activation (as depicted in Figure 2). For the preparation of body homogenates, 5 bodies without heads were utilized, whereas head homogenates were prepared from 32 heads. The samples were processed following the same protocol employed for TMPyP3 quantification. The concentration of H_2_O_2_ in the body and head homogenates was determined using a calibration curve for dihydroethidium (DHE, ≥95%, Sigma Aldrich) with known H_2_O_2_ concentrations [21]. The reaction mixture comprised 1 × PBS (pH 7.4), 10 μM DHE, and 5 μL of homogenate, with a final volume of 200 µL. The microplate containing the samples was then incubated for 30 min at 37 °C in the absence of light. H_2_O_2_ in the homogenates was determined using a Tecan Infinite Pro200 microplate reader with an λ_ex_ 480 nm and λ_em_ at 625 nm.

### 2.6. Negative Geotaxis

To evaluate the potential neurotoxic impact of light-activated TMPyP3 on adult male *Drosophila*, we conducted behavioral assessments using the negative geotaxis test. Negative geotaxis evaluates the rate of vertical climbing following the disturbance of flies to the bottom of a vial. This method is frequently employed to gauge the effects of aging and neurodegeneration [22]. We quantified negative geotaxis by determining the percentage of flies that ascended beyond the midpoint of the vertical wall of the vial, situated at a height of 10 cm, five seconds after being dislodged to the vial’s base. Flies were acclimated to the test vials for 30 min before experimentation. The procedure entailed tapping the vials three times on a firm surface and capturing images five seconds thereafter. Through image analysis, we computed the proportion of flies that traversed the 10 cm vial midline within five-second intervals. This process was repeated five times, with one-minute intervals between measurements. We evaluated 3–5-day-old flies that were exposed to 40 μg of TMPyP3 in M3 medium for either 180 min or 24 h and subjected to illumination at 40 mW/cm^2^ for durations of 10 or 20 min. Subsequently, these flies were allowed to age for seven days on M3 in constant darkness before behavioral testing.

### 2.7. Data Analysis and Statistics

Biochemical and behavioral data were computed by MS Excel, while statistical analyses and visualizations were conducted in Prism 10.0.3 (GraphPad, La Jolla, CA, USA). Treatment disparities were assessed employing either unpaired t-tests or one-way ANOVA followed by Tukey’s multiple comparisons test, contingent on the dataset. The normality of the data was evaluated using Bartlett’s test or Brown–Forsythe’s test. Significance was acknowledged for *p*-values < 0.05.

## 3. Results

### 3.1. Starvation in Combination with a Short Duration of Feeding on the Yeast Free Media Leads to Maximal TMPyP3 Ingestion

The chemical stability of TMPyP3 was tested in three different *Drosophila* nutrient media after 24 h of incubation (Figure 3A). In the yeast-free M3 medium, there is no intensity decrease in the fluorescence emission spectrum of TMPyP3, while there is a significant decrease in the M1 and M2 media (Figure 3A). Based on these results, in all the subsequent experiments, the oral administration of TMPyP3 to adult flies was carried out using M3 medium.

To optimize the feeding condition that would result in the highest TMPyP3 ingestion in the whole body, we starved the flies for 16 h before feeding for 30, 90 or 180 min, and compared that to the amount of TMPyP3 in flies that were fed for 24 h without starvation. In the starved groups, the highest ingested concentration of TMPyP3 in the whole-body homogenates was in the group that was fed for 180 min (Figure 2B). Interestingly, providing M3 medium with TMPyP3 *ad libitum* for 24 h resulted in a lower concentration of TMPyP3 in the whole-body relative to the starved groups. Based on this outcome, we formulated our standard feeding protocol to consist of 16 h of starvation followed by 180 min of feeding on M3 medium enriched with 40 µM TMPyP3.

### 3.2. TMPyP3 Retention in the Body Causes a Light-Independent Decrease in the Levels of H_2_O_2_

Because we optimized the protocol for the maximum concentration of ingested TMPyP3 using the extracts of the whole flies, we were interested if there was a difference in the amount of TMPyP3 absorbed and retained in the body in comparison to the head. To provide sufficient time for food to be metabolized and for TMPyP3 concentration to reflect the amount present in different body organs, we measured the concentration of TMPyP3 in the bodies of flies one and seven days after TMPyP3 treatment. To avoid the potential effect of light on the activation of TMPyP3, following the feeding, the flies were kept inconstant darkness on M3 medium without TMPyP3.

Additionally, to see if the photoactivation by red light (λ = 645 nm, 40 mW/cm^2^) for 10 min affects TMPyP3 concentration and the subsequent production of hydrogen peroxide (H_2_O_2_), flies either received light treatment (L) or not (NL) before remaining for one and seven days in constant darkness. Experimental groups were starved for 16 h, fed for 180 min on M3 medium containing 40 µM of TMPyP3, exposed to red light (L) or not (NL), and then kept for one or seven days in darkness on M3 medium. Control groups were not fed TMPyP3 and then were either exposed to light (L) or not (NL), followed by one or seven days of darkness.

In the body, TMPyP3 is present one and seven days post-feeding. TMPyP3 concentration is decreased seven days post-treatment; however, more than 50% of the TMPyP3 amount is still present, suggesting slow clearance from the body (Figure 4A,B). Red-light treatment did not significantly change the TMPyP3 concentration in the body relative to flies that were fed TMPyP3 but not exposed to red light, and the effect persists for seven days (Figure 4A,B). One day after the red-light exposure, the amount of H_2_O_2_ increased (Figure 4C), but after seven days, the amount of H_2_O_2_ was significantly lower in both the NL and L groups compared to the controls (Figure 4D). We also detected an increase in H_2_O_2_ concentration in L versus NL controls that were not fed TMPyP3 (Figure 4D).

### 3.3. Retention of TMPyP3 in the Head Decreases the H_2_O_2_ Concentration after Seven Days

To examine the potential neurotoxic effect of TMPyP3 on *Drosophila*, we quantified TMPyP3 in the head homogenates and measured the concentration of H_2_O_2_.

TMPyP3 was present in the head one day after the treatment, and this did not change significantly after seven days (Figure 5A,B). Similar to the body, photoactivation did not influence the concentration of TMPyP3 in the head after one or seven days (Figure 5A,B). However, the photoactivation of TMPyP3 led to an increased production of H_2_O_2_ in head extracts measured one day after the exposure (Figure 5C), but there was a significant decrease relative to the controls after seven days (Figure 5D). A similar decrease in the H_2_O_2_ amount was previously observed in the body homogenates (Figure 4D).

### 3.4. Retention of TMPyP3 in the Head Decreases Negative Geotaxsis

Negative geotaxis measures the speed of vertical climbing and is commonly used to assess the effects of ageing and neurodegeneration [22]. Since TMPyP3 was present in the head and body at least seven days after the treatment and changed the amount of H_2_O_2_, we were interested to see if that affected a simple motor function, such as vertical climbing ability. To determine if there are long-term behavioral effects of TMPyP3 feeding, we quantified the ability for vertical climbing in groups of flies where we varied the duration of feeding on M3 media containing TMPyP3, starvation and the duration of photoactivation.

We found that the negative geotaxis seven days after TMPyP3 treatment depended on the length of TMPyP3 feeding but not on the duration of photoactivation (Figure 6). Decreased negative geotaxis was observed in the group starved for 16 h, fed TMPyP3 for 180 min and exposed to red light for 10 min. This result confirms that our optimal feeding protocol, which led to the highest concentration of TMPyP3 in the whole body (Figure 3B), also led to the largest effect on the negative geotaxis seven days after photoactivation. We cannot confirm if the duration of photoactivation had a significant effect because there was no significant difference between groups photoactivated for 10 or 20 min. However, both the 10 and 20 min photoactivation groups in the optimal protocol group had a significantly decreased negative geotaxis compared to their respective controls (10 or 20 min photoactivation in flies fed for 24 h without prior starvation). Because negative geotaxis was measured seven days post-TMPyP3 treatment, it correlates with the presence of TMPyP3 in the head and the body and the effect on lowering the H_2_O_2_ concentration.

## 4. Discussion

To determine the systemic effects of photoactivated TMPyP3 on healthy adult male flies, we optimized the administration protocol and measured the effect of the ingested and retained TMPyP3 on the redox indicator—H_2_O_2_—and on the behavioral marker of neurodegeneration—the negative geotaxis test. We identified the optimal feeding media and defined the administration protocol to reproducibly achieve a high concentration of ingested TMPyP3 in the body. TMPyP3 is retained in the body and in the head up to seven days post-administration, suggesting absorption in different organs and tissues. TMPyP3 photoactivation did not increase H_2_O_2_ concentration in the flies’ heads and bodies, except when measured one day post-treatment in the head. Interestingly, TMPyP3-treated flies had a consistently decreased H_2_O_2_ concentration after seven days, irrespective of the light treatment. Finally, there was a significant decrease in the climbing ability of the TMPyP3-treated flies, suggesting a long-term systemic effect.

We show that the optimal feeding media for ingesting TMPyP3 should not contain yeast. The chemical stability and concentration of TMPyP3 in the medium varied depending on the presence of yeast in the nutrient medium, and there was a significantly lower fluorescence intensity of TMPyP3 in two media containing inactivated yeast (M1 and M2) compared to the yeast-free media (M3). This might be supported by findings that yeast can digest porphyrins [23,24]. However, since we used inactivated yeast, we speculate that there are electrostatic interactions between yeast proteins and TMPyP3 [25]. Further testing will be required to understand the mechanism of interaction between the inactivated yeast, which is a common *Drosophila* food supplement, and TMPyP3.

The highest ingested concentration of TMPyP3 in the whole-body homogenate was achieved by the combination of starvation and restricted feeding on the yeast-free M3 media. Different durations of starvation are often employed as a pre-treatment to increase the amount of ingested food per unit of time. Additionally, starvation reduces the rate of defecation, thereby prolonging the retention of food in the digestive system [26,27], which could explain the higher absorption of TMPyP3 after starvation. Food retention also prolongs the available time for the absorption of substances present in the food [28,29]. The absorption of TMPyP3 in the digestive tract is likely pH dependent. The cationic nature of TMPyP3 favors the passage through the positively charged portions of the midgut of the *Drosophila* digestive system, resulting in the increased absorption in the negatively charged area [26].

TMPyP3 concentration in the body and head after one and seven days suggests rapid accumulation and slow clearance, particularly from the head. One day after feeding, TMPyP3 is likely present not only in the digestive tract but in other organs as well. A related PA, ruthenium(II) polypyridyl-substituted porphyrin metalated with zinc(II) (named Complex I), was tested using *Drosophila’s* larvae [19]. After incubating dissected larval tissue for 30 min in a medium containing Complex I, its presence was detected in the cytoplasm and nucleus of different tissues, including the gut and larval brain. A similar result was obtained when larvae were fed for 48 h on the media containing Complex I [19]. Although we quantified TMPyP3 in the homogenates, we suspect that after one day, TMPyP3 could be present in the cells of the digestive tract and in the hemolymph—the equivalent of the blood in humans, which bathes *Drosophila’s* organs and tissues [30]. After seven days, we suspect that TMPyP3 has been transported via the hemolymph to nearly all organs and tissues. Decreased concentration after seven days, relative to the first day, could be due to the activation of cellular mechanisms involved in metabolism, detoxification and excretion [31].

The presence of TMPyP3 in the head after just one day likely stems from transport to the brain, as was reported for the larval brain [19]. In addition to being hydrophilic and soluble in polar solvents, TMPyP3 can pass through the *Drosophila* blood–brain barrier (BBB). *Drosophila’s* BBB is formed of glial cells [32,33,34], in contrast to the mammalian, which consists of polarized endothelial cells [35,36,37]. From hemolymph, only gases and small lipophilic molecules can cross the fly’s BBB plasma membrane by passive diffusion. The similar head concentration of TMPyP3 over the course of seven days relative to the body reflects the difference in cellular mechanisms involved in the clearance and metabolism of foreign molecules.

We measured H_2_O_2_ to determine if TMPyP3 alone or after photoactivation led to the production of H_2_O_2_. Photoactivation increased H_2_O_2_ concentration only in the head extracts one day post-treatment. The lack of a light-induced increase in H_2_O_2_ concentration in the body after seven days could be due to several reasons. One is that the major increase in H_2_O_2_ concentration occurs shortly after the photoactivation, and H_2_O_2_ levels normalize after one day. The second is that a better indicator of the oxidative stress or the endogenous antioxidative response that persists for more than one day could be the superoxide dismutase (SOD), catalase or other enzymes in the glutathione pathway. However, both explanations can be partially refuted by the fact that photoactivation by light did increase H_2_O_2_ in the head. The third explanation pertains to the location of TMPyP3 in the body. *Drosophila’s* exoskeleton is transparent, allowing for light to penetrate to the brain but not to body organs that are further removed from the exoskeleton. Crystallographic studies performed on human serum albumin reveal distinct binding sites for porphyrins, which can bind non-covalently through positive and negative ionic anchoring or docking into large hydrophobic pockets [38,39,40]. These interactions can be altered due to changes in ionic strength, pH or polarity, but they can interfere with free amino acid residuals [41]. Since, in the body, the homogenates we have measured produce a lower TMPyP3 concentration after seven days than after one day, a possible explanation could be that TMPyP3 is inactive by forming π–π and cation–π interactions in a protein–porphyrin complex.

The depth of light penetration provides the most plausible explanation for the light-induced increase in H_2_O_2_ concentration in the head and the absence of the increase in the body. Underneath the fly’s head exoskeleton is a single-cell deep layer that provides a chemical and physical barrier and consists of perineuronal glia and macrophages. Beneath them are subperineuronal glia that constitute the BBB [42]. The relative proximity of glia to the head surface might be sufficient for the light penetration through the exoskeleton and activation of TMPyP3, leading to the measurable increase in H_2_O_2_ concentration. This would also indicate that TMPyP3 was transported into one of the glial layers. The H_2_O_2_ elevation is not evident after seven days because it has been metabolized by catalase or other antioxidative enzymes. However, the level of TMPyP3 in the head remained similar over seven days, indicating that it has not been metabolized or excreted. TMPyP3 might persist in the surface glia, or in neurons or other glial types that reside deeper in the brain, because the BBB efficiently uptakes and secretes ions and metabolites [43].

The fact that, after seven days, the concentration of H_2_O_2_ decreases relative to respective controls in the head and body homogenates independent of the photoactivation suggests that the long-term presence of TMPyP3 in different cells interferes with redox regulation. In the head, where light photoactivation led to an increase in H_2_O_2_, this could activate the endogenous antioxidant defense, such as a long-term increase in catalase activity that led to lower long-term H_2_O_2_ levels. Similar results have been observed in wild-type mice, where the injection of protoporphyrin-IX led to an increase in the catalase activity followed by an increase in the medium life span by 26% [44]. In the wild-type *Drosophila*, protoporphyrin-IX increased the medium life span, but not in the strain that was deficient in catalase activity, implying the importance of catalase activity for life extension induced by protoporphyrin-IX [18]. These life span-extension results lend support to the theory of ageing that proposes the damaging effects of accumulated free radicals.

However, H_2_O_2_ levels in the cell are tightly controlled because the concentration and the spatiotemporal localization of H_2_O_2_ can have negative effects on cell functioning when it is above or below the optimal physiological level [45,46,47]. For example, the overexpression of catalase in *Drosophila* does not lead to the extension of the life span, although that is predicted due to its role in the metabolism of H_2_O_2_. At the physiological level, H_2_O_2_ has an important role as a second messenger that can oxidize the redox-sensitive thiol group in cysteine-containing proteins and consequently modify the activity of transcription factors, tyrosine phosphatases and mitogen-activated protein kinases [48]. The significant decrease in H_2_O_2_ that we observed seven days after the treatment suggests that the cellular presence of TMPyP3 interfered with proteins that are the source of H_2_O_2_ (NADPH oxidase and superoxide dismutase) or the antioxidant defense (catalase, glutathione peroxidases or peroxiredoxins). However, another possibility is that TMPyP3 led to a decreased H_2_O_2_ level due to the interaction of porphyrin with the cellular iron. Porphyrins are known to chelate metal ions, especially those with ionic radii of ~70 pm such as Cu(II), Fe(II), Ni(II) and Zn(II) ions [49]. The insertion of a metal ion is a very slow process, but the porphyrin complexes formed with the mentioned transition metal ions are stable [50]. Complexes of TMPyP4 and TMPyP3 with biologically relevant ions, such as iron and zinc ions, have been synthesized [51,52], so it is possible to expect the interaction of these free base porphyrins with the same metal ions in living systems, especially over a longer period of time. For example, it has been shown that the insertion of Zn(II) ions into TMPyP4 molecules is not only possible at neutral and slightly acidic pH but in the presence of biopolymers—this process is even more effective [53].

Head extracts do not show a reduction in the TMPyP3 concentration, so it is possible that TMPyP3 may influence intracellular iron. The presence of intracellular iron has a strong impact on cellular redox status. Within the cell, iron mediates electron transfer reactions by switching between two states: ferrous(II) and ferric(III) iron. This mechanism can produce distinct levels of ROS, which activate calcium-mediated basal synaptic transmission and long-term potentiation [54], but this may catalyze a Fenton reaction, which generates ROS in high amounts, which is detrimental for cellular function, and eventually, survival [55]. Interestingly, we have not observed the chelation of metal ions in the TMPyP3 cavity of biologically relevant ions in different solvents, so the lowered values could be due to the influence on endogenous redox genes and proteins since the effect of H_2_O_2_ is dual—significantly elevated levels can be toxic, while moderate and regulated levels participate in signaling. Redox behavior is described for TMPyP4 as a compound oxidized on a glassy carbon electrode [56] and in the context of its known ability to form interactions with nucleic acids, which is also known for TMPyP3 [57]. As for the antioxidant effect, it is known that porphyrins with a TMPyP3 and TMPyP4 structure, chelated with manganese Mn(III) ions, can act as superoxide dismutase mimics [58,59], while in free-base porphyrins, no antioxidant effect was observed.

Our results of impaired negative geotaxis provide support for the importance of tightly controlled physiological levels of H_2_O_2_ required for optimal cellular functioning. The TMPyP3-induced decrease in H_2_O_2_ correlated with a decrease in climbing ability. Based on the importance of H_2_O_2_ for cellular signaling, it is possible that the decreased H_2_O_2_ levels impaired cellular functioning in the cells that control climbing ability. Since the H_2_O_2_ decrease occurred in the head and the body, we cannot with certainty claim that decreased negative geotaxis is due to neurodegeneration, although negative geotaxis is sensitive to the loss of dopaminergic neurons in the brain [60,61]. Alternatively, or in addition, decreased H_2_O_2_ could impair the processes at the neuromuscular junction [46] or in the muscles involved in climbing response [62].

## 5. Conclusions

We have identified an optimal administration protocol for TMPyP3 in *Drosophila*, which persisted for up to seven days post-administration. Our study describes the advantages of using an invertebrate model organism to develop and test the effects of novel PA on redox regulation and behavioral phenotype. In agreement with human studies, it shows that the effectiveness of photosensitizing agent to induce H_2_O_2_ depends on the PA being close to the surface of the body. The ease of quantifying behavioral consequences of PDT in *Drosophila* can expedite the screening of new PAs that could have long-term local or systemic effects.

## Figures and Tables

**Figure 1 biotech-13-00023-f001:**
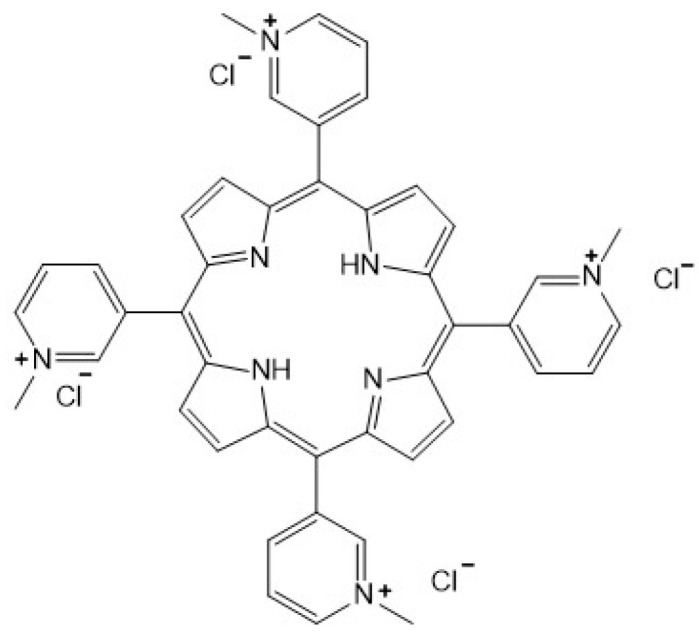
Structure of 5,10,15,20-tetrakis(*N*-methylpyridinium-3-yl)porphyrin (TMPyP3).

**Figure 2 biotech-13-00023-f002:**
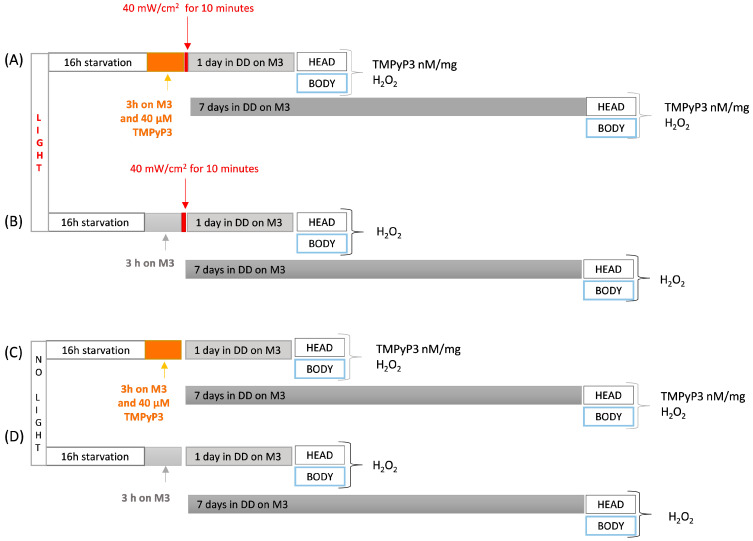
Optimized protocol for testing TMPyP3 retention and the levels of hydrogen peroxide (H_2_O_2_). In all experiments, 3–5-day-old adult males were used, which were 16 h starved. (**A**) Flies were fed for three hours with 40 μM of TMPyP3 on M3 in the dark and illuminated with red light (λ = 645 nm, 40 mW/cm^2^) for ten minutes, after which they were transferred to the M3 and left in constant darkness (DD) for one or seven days. Samples of heads and body were collected and TMPyP3 absorption and H_2_O_2_ concentration were measured. (**B**) Flies were fed for three hours on plain M3 in the dark and illuminated with red light (λ = 645 nm, 40 mW/cm^2^) for ten minutes, after which they were transferred to the fresh M3 and left under constant darkness (DD) conditions for one or seven days. Samples of heads and body were collected and H_2_O_2_ concentration was measured. (**C**) Same as in (**A**) but without light treatment (**D**) Same as in (**B**) but without light treatment.

**Figure 3 biotech-13-00023-f003:**
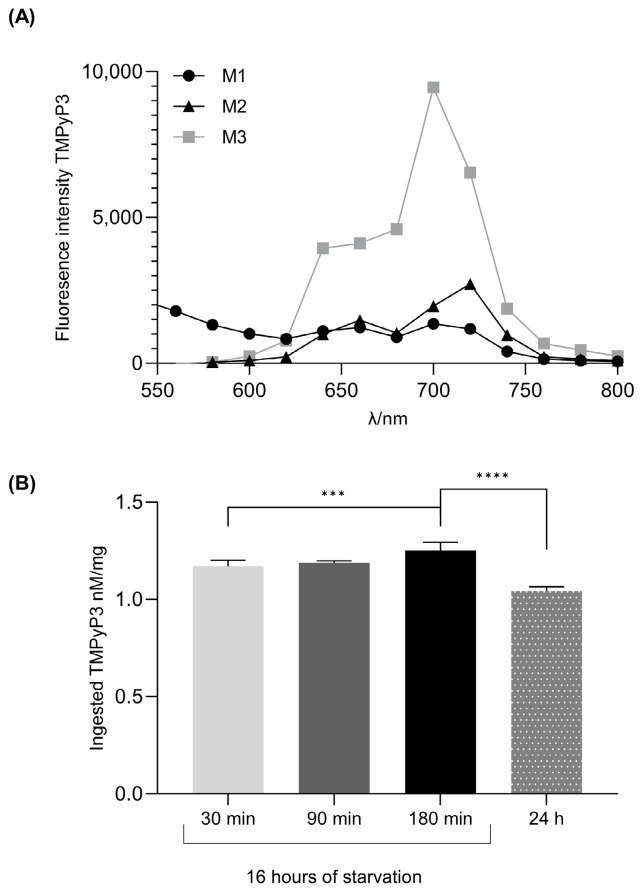
Optimization of nutrition media and oral TMPyP3 administration. (**A**) Chemical stability of TMPyP3 in different food media. Fluorescence emission spectra of TMPyP3 in range 550–800 nm using excitation wavelength of 420 nm in M1 (standard food media), M2 (agar + sugar + yeast) and M3 (agar + sugar) after 24 h. The fluorescence emission spectra of TMPyP3 are shown as the difference between the fluorescence spectrum of TMPyP3 in the medium and the spectrum of the medium alone. All samples were recorded as duplicates and plotted as mean values. (**B**) Absorbed nM of TMPyP3 per milligram of *Drosophila* whole-body homogenate after the administration of 40 µM TMPyP3 in the M3 medium. Groups of flies (*n* = 15 in each group) were starved for 16 h and fed for 30, 90 and 180 min with 40 μM of TMPyP3 on the M3 medium or fed 24 h without starvation. Results are the mean value of triplicate ± standard error mean (SEM). The statistical significance was determined using a one-way ANOVA test, with Bonferroni correction. Significant values: ***: *p* < 0.05 and ****: *p* < 0.01.

**Figure 4 biotech-13-00023-f004:**
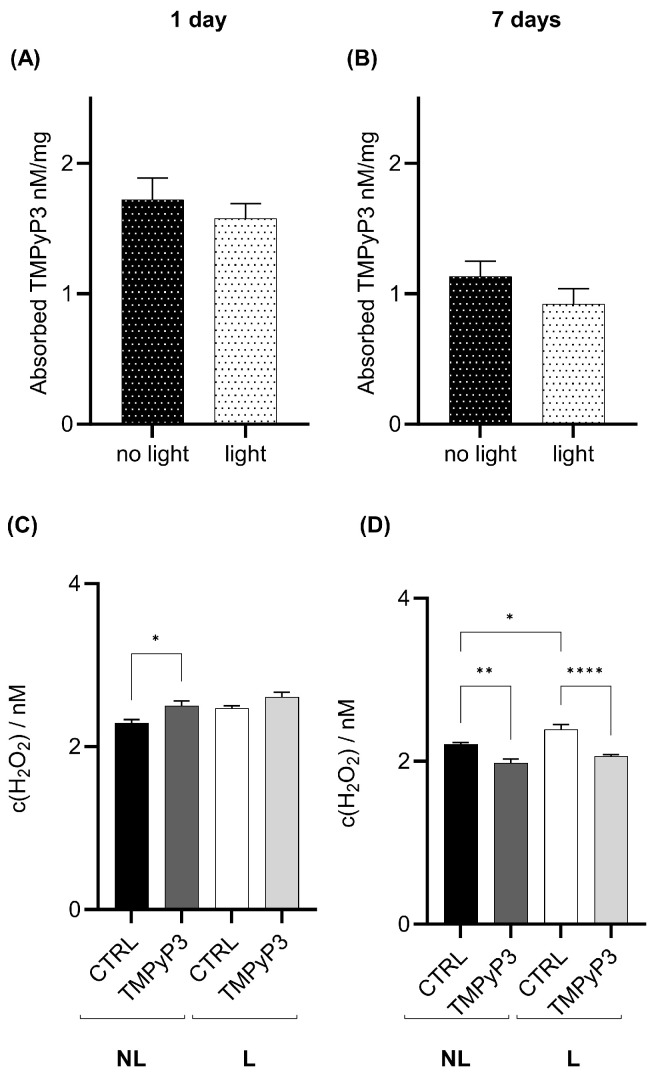
TMPyP3 present in the body homogenates seven days after feeding decreases the concentration of H_2_O_2_. Samples of flies’ headless bodies were collected one (**A**,**C**), and seven days (**B**,**D**) after the oral administration of 40 µM of TMPyP3 for 180 min in M3 medium and illumination with red light (645 nm, 40 mW/cm^2^) for 10 min, (L) group, or were not illuminated—no light (NL). (**A**,**B**) TMPyP3 in nM per milligram of *Drosophila* body. (**C**,**D**) H_2_O_2_ concentration determined using calibration curve for dihydroethidium (DHE). Controls (CTRL) flies underwent the same procedure (L or NL) but without TMPyP3. All measurements were performed in triplicates (*n* = 9). One-Way ANOVA with Tukey’s multiple comparison post hoc test. Significant values in all panels: *: *p* < 0.05, **: *p* < 0.01, ****: *p* < 0.001.

**Figure 5 biotech-13-00023-f005:**
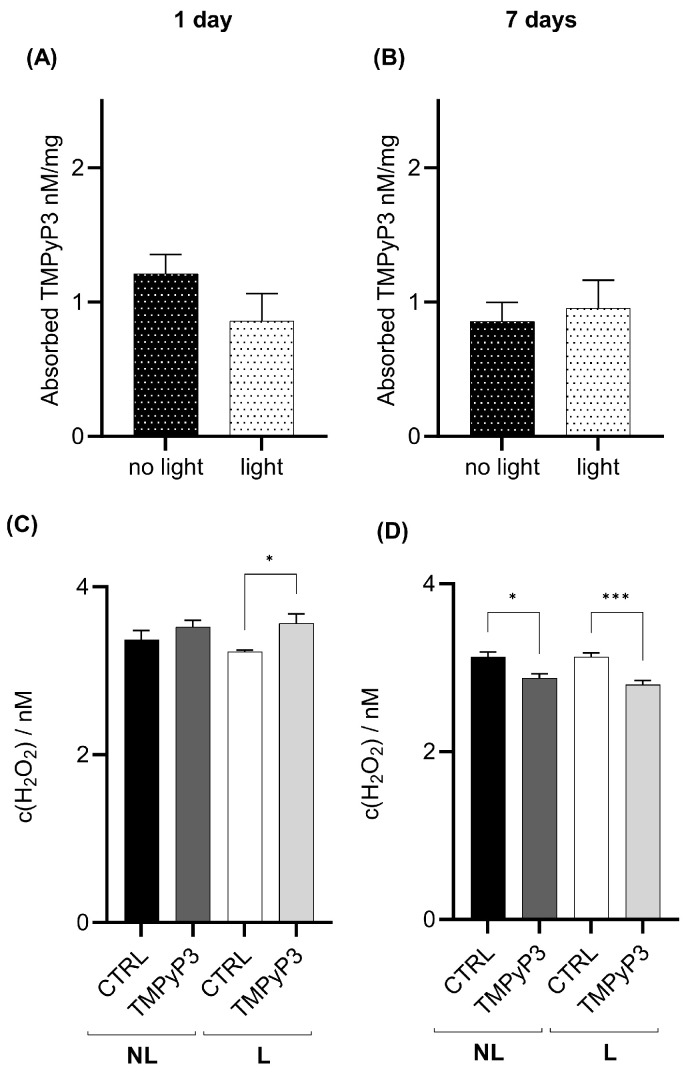
In the head homogenates, the concentration of TMPyP3 does not change during seven days and decreases the concentration of hydrogen peroxide. Samples of flies’ heads were collected one (**A**,**C**), and seven days (**B**,**D**) after oral administration of 40 µM of TMPyP3 for 180 min through the M3 medium and with illumination with red light (645 nm, 40 mW/cm^2^) for 10 min (L) or without illumination—no light (NL). (**A**,**B**) TMPyP3 in nM per milligram in head homogenate. (**C**,**D**) Hydrogen peroxide (H_2_O_2_) concentration determined using calibration curve for dihydroethidium (DHE). Controls (CTRL) flies underwent the same procedure (L or NL) without TMPyP3. All measurements were performed in triplicates (*n* = 9). *: *p* < 0.05; ***: *p* < 0.01 using One-Way ANOVA with Tukey’s multiple comparison post hoc test.

**Figure 6 biotech-13-00023-f006:**
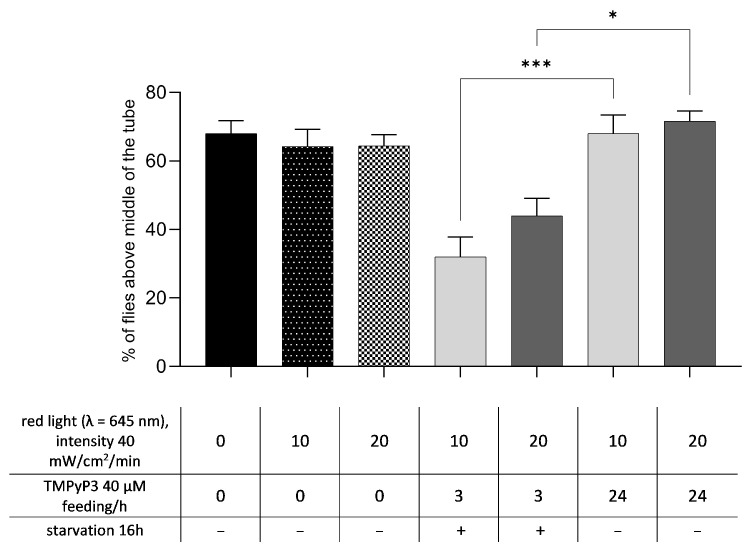
Photoactivated TMPyP3 decreases negative geotaxis. Ability of flies to climb vertical surface depending on the starvation period prior to oral administration of TMPyP3, the duration of the feeding with TMPyP3 and the photoactivation with red light (λ = 645 nm; 40 mW/cm^2^). The results are presented as the mean value of the measurements in triplicate ± SEM (*n* = 50 per treatment, performed in 5 replicas). One-way ANOVA test with Bonferroni correction, statistical significance *: *p* < 0.05; ***: *p* < 0.01.

## Data Availability

Raw data from the study can be found here https://doi.org/10.5061/dryad.bk3j9kdm0 accessed on 18 April 2024.

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
