# Peer review of "Systemic Effects of Photoactivated 5,10,15,20-tetrakis(N-methylpyridinium-3-yl) Porphyrin on Healthy Drosophila melanogaster"

_biotech, 2024, doi:10.3390/biotech13030023_

Round 1
Reviewer 1 Report
Comments and Suggestions for Authors
The study by Dr Ana Filošević Vujnović and co-aothors proposes
Drosophila as a test system for photosensitising agents.
These types of studies are to be encouraged, as some pre-clinical
evidence is gathered while raising less ethical concerns compared to
studies on mammals.
A few more citations may be in order for the first paragraph of the introduction, especially for the general topic of photodynamic therapy (e.g., Lim, Heeo, et al., Cancer Lett 2013).
The text is well written, there are few instances where the Ms needs
minor revisions to eliminate repetitions - e.g. (type I reaction) react - or vague sentences "influence cell viability".
In the phrase:
"However, efficiency and potential toxicity of different PAs using Drosophila has been limited"
--> do the authors mean <<the testing of efficiency and toxicity>>?
The Introduction is rather long.
Error bars in figures are often quite high compared to differences in data, (especially in Fig4, Fig 5), yet
significant differences are corrrectly indicated.
The conclusions are supported by this extensive study, however the following
point needs to be made:
"We have identified an optimal administration protocol for TMPyP3"-->
"We have identified an optimal administration protocol for TMPyP3 in Drosophila.."
Overall, a good study, I recommend publication once the suggestions for minor modifications are taken into account.
English is mostly fine.
Reviewer 2 Report
Comments and Suggestions for Authors
I found this paper interesting, well-designed, and properly written. Nevertheless, I have some minor suggestions for the authors.
- Firstly, it was unclear to me what the reason was for such food protocols. This was later described in the paper, but I believe it should be clarified more at the beginning.
- I believe that Section 2.3 is quite chaotic and should be rewritten to be more straightforward. Perhaps a figure that graphically shows the feeding protocol could help. Additionally, this section should be divided into two parts: one for the feed protocol (e.g., 2.3.1) and another for measurements (e.g., 2.3.2).
- On Figure 5, I do not know what the abbreviation "DD" stands for.
- Were the ages of the flies the same for all experiments? Were they freshly hatched or older specimens? Because antioxidation mechanisms can change during the aging process, I believe this needs to be taken into consideration in the discussion and experiment description sections.
- The authors mention a starvation procedure in the experiment description. This makes a lot of sense, but starvation can also lead to an increase in oxidation/antioxidation processes and changes in H2O2 levels. This should also be taken into consideration in the discussion part. I think the authors should check the H2O2 level in non-starved specimens as a baseline for these studies, just to ensure that starvation did not change the level of oxidative stress.
- The authors mention in the discussion that glutathione levels may change the response to H2O2 production. I believe it would be good to measure the changes in glutathione levels as an added value to this experiment. It is a relatively straightforward test that could answer some questions raised in the discussion.
Round 2
Reviewer 2 Report
Comments and Suggestions for Authors
After changes made by the authors, I have no further comments. The paper is well designed and well written.
Congratulations to the authors and good luck in future studies.